# Compared with Cotrimoxazole Nitroxoline Seems to Be a Better Option for the Treatment and Prophylaxis of Urinary Tract Infections Caused by Multidrug-Resistant Uropathogens: An In Vitro Study

**DOI:** 10.3390/antibiotics10060645

**Published:** 2021-05-28

**Authors:** Ulrich Dobrindt, Haleluya T. Wami, Torsten Schmidt-Wieland, Daniela Bertsch, Klaus Oberdorfer, Herbert Hof

**Affiliations:** 1Institut für Hygiene, Universitätsklinikum Münster, 48149 Münster, Germany; Haleluya.Wami@ukmuenster.de; 2MVZ Labor Limbach und Kollegen, Im Breitspiel 16, 69126 Heidelberg, Germany; torsten.schmidt-wieland@labor-limbach.de (T.S.-W.); Daniela.Bertsch@labor-limbach.de (D.B.); Klaus.Oberdorfer@labor-limbach.de (K.O.); Herbert.Hof@labor-limbach.de (H.H.)

**Keywords:** parallel resistance, resistance plasmid, horizontal gene transfer, prophylaxis of uncomplicated UTI

## Abstract

The resistance of uropathogens to various antibiotics is increasing, but nitroxoline remains active in vitro against some relevant multidrug resistant uropathogenic bacteria. *E. coli* strains, which are among the most common uropathogens, are unanimously susceptible. Thus, nitroxoline is an option for the therapy of urinary tract infections caused by multiresistant bacteria. Since nitroxoline is active against bacteria in biofilms, it will also be effective in patients with indwelling catheters or foreign bodies in the urinary tract. Cotrimoxazole, on the other hand, which, in principle, can also act on bacteria in biofilms, is frequently inactive against multiresistant uropathogens. Based on phenotypic resistance data from a large number of urine isolates, structural characterisation of an MDR plasmid of a recent ST131 uropathogenic *E. coli* isolate, and publicly available genomic data of resistant enterobacteria, we show that nitroxoline could be used instead of cotrimoxazole for intervention against MDR uropathogens. Particularly in uropathogenic *E. coli*, but also in other enterobacterial uropathogens, the frequent parallel resistance to different antibiotics due to the accumulation of multiple antibiotic resistance determinants on mobile genetic elements argues for greater consideration of nitroxoline in the treatment of uncomplicated urinary tract infections.

## 1. Introduction

The prevalence of multidrug-resistant (MDR) uropathogens, i.e., bacteria resistant to one antibiotic of at least three different antibiotic classes [1] is increasing worldwide among both in-patients as well as out-patients with urinary tract infections (UTIs) [2,3,4,5,6,7]. Patients with indwelling catheters are particularly at risk, and catheter-associated UTIs, as one of the most common nosocomial infections, favour infection by MDR bacteria [8,9,10,11].

UTIs with uropathogens resistant to 3rd generation cephalosporins and quinolones are associated with a poor outcome because initial antibiotic therapy with the first-line antibiotics usually recommended for the treatment of UTIs is increasingly inadequate to ensure efficient elimination of these pathogens [12]. Treatment with fosfomycin, pivmecillinam or nitrofurantoin is currently recommended for first-line treatment of mild and uncomplicated UTIs [13,14,15,16]. Carbapenems are used as empirical therapy of symptomatic UTIs caused by MDR bacteria [17,18]. However, the steadily increasing prevalence of ESBL-producing uropathogens requires an increased clinical use of carbapenems [19], and as a consequence, carbapenem resistance is also spreading [20]. In order to reduce the prescription of carbapenems [21], the question arises of which oral antibiotics, besides fosfomycin, nitrofurantoin and cotrimoxazole, are still available for the management (and prophylaxis) of these difficult-to-treat pathogens [21,22,23]. Cotrimoxazole, the combination of trimethoprim and sulfamethoxazole, inhibits the bacterial biosynthesis of tetrahydrofolic acid, which is essential for the formation of thymidine and purine bases, and thus for the synthesis of DNA. Since this drug exerts its antimicrobial activity via a mechanism unrelated to that of β-lactams and quinolones, it is also recommended as a second-choice drug, at least when the local resistance pattern for *E. coli* in the treating hospital or in the closer geographical vicinity is <20% [13,21,24].

Since nitroxoline is active against a broad spectrum of different Gram-negative and Gram-positive bacteria as well as against yeasts isolated from urine samples from patients [24,25,26,27,28,29], it can also be considered as another possible treatment for UTIs. In the new German guidelines for uncomplicated UTIs, nitroxoline is recommended as one of the drugs of primary choice for calculated therapy of UTIs in adults [24,30]. Furthermore, this drug can be used for long-lasting prophylaxis of recurrent UTIs. Indeed, the most common causes of UTIs such as *E. coli* [27,31] are generally susceptible to nitroxoline [26]. This drug is even active against bacteria in biofilms [27,31]. Other causative agents of UTIs, such as *Pseudomonas aeruginosa* and *Enterococcus faecalis* are, however, generally resistant [25,26]. Oral application results in high concentrations of nitroxoline and its active metabolite nitroxoline sulfate in urine, which is a prerequisite for successful therapy of uropathogens [32]. Although sound data on the pharmacology of this old drug are still lacking [33], the drug has been shown to be well tolerated in a meta-analysis [34]. Only mild adverse drug reactions including gastrointestinal disorders and allergic reactions, but no marked alterations of the composition of the fecal microbiome, have been observed [30,34,35].

The increasing risk of antimicrobial resistance is a worldwide concern, not only because of the limited treatment options, but also because of the rapid spread of mobile genetic elements (MGEs), including transposons, integrons, plasmids and bacteriophages, which carry antimicrobial resistance genes [36,37]. The frequent co-residence of several resistance determinants on plasmids or integrons promotes co-selection of multiple antibiotic resistances, and thus the spread of multiple antibiotic resistance among bacteria. Integrons are of great clinical importance, because they most often contribute to the accumulation of multiple gene cassettes coding for antibiotic and disinfectant resistance [38,39,40]. Consequently, exposure to one antibiotic is sufficient to promote the expression of unrelated, but co-resident AMR determinants on an MGE, and eventually the spread of the co-resident unrelated resistance genes. Parallel resistance to other antibiotics increases the risk for co-selection, maintenance and transmission of MDR plasmids and/or MDR uropathogens.

Using in vitro susceptibility tests of bacterial isolates from urine samples, we evaluated nitroxoline sensitivity among *Enterobacterales* isolated from urine. We also analysed the complete genome sequence of a randomly selected multiresistant *E. coli* urine isolate identified from our susceptibility tests to identify which AMR determinants associated with MGEs were present in the strain. In this manner, we assessed and exemplarily described the genomic basis for the parallel resistance of *E. coli* urine isolates to many antibiotics. Our results indicate that the current guidelines for treatment of symptomatic UTI may fuel the selection of MDR uropathogens. In contrast, our results also suggest that the use of nitroxoline is associated with a low risk of resistance and the co-selection of other resistances.

## 2. Results

### 2.1. Prevalence of MDR among E. coli Isolates from Urine Samples

A total of 140,648 urine samples were analyzed in 2018 for the presence of uropathogens. If the number of colony-forming units in a sample was greater than 10^4^/mL and there were no more than three different pathogens in this urine sample, the different urine isolates were declared relevant and included in the study. Altogether, 35,390 *E. coli* isolates, 6227 *Klebsiella pneumoniae* isolates and 5088 *Proteus mirabilis* isolates were collected from these urine samples.

Out of 35,390 *E. coli* urinary isolates 763 strains were resistant to 3rd generation cephalosporins and 2628 strains were MDR, i.e., resistant to penicillins, 3rd generation cephalosporins as well as to ciprofloxacin (Figure 1). Of the 6227 *K. pneumoniae* isolates, 112 strains were resistant to 3rd generation cephalosporins, whereas 467 isolates were MDR. In case of the 5088 *P. mirabilis* isolates, one strain was resistant to 3rd generation cephalosporins and 41 strains were MDR.

The rate of MDR or resistance to 3rd generation cephalosporins among the *E. coli* and *K. pneumoniae* urine isolates was significantly higher than among *P. mirabilis* isolates (two-sided Fisher’s exact test, *p* < 0.0001). *E. coli* and *K. pneumoniae* urine isolates differed not significantly in the prevalence of MDR or resistance to 3rd generation cephalosporins, respectively (two-sided Fisher’s exact test, *p* > 0.999 and *p* = 0.0709).

### 2.2. Overall Activity of Nitroxoline against Multiresistant Uropathogens

The susceptibility to nitroxoline was tested for a subset of the bacterial strains we isolated from urine. The number of strains we could test was limited, because this antimicrobial is not part of the routine commercial test kits and has to be tested manually. Among 600 susceptible *E. coli* isolates (i.e., strains susceptible to penicillins, 3rd generation cephalosporins and quinolones) examined, only three were resistant to nitroxoline. None of the 88 MDR *E. coli* isolates tested displayed resistance to nitroxoline (Figure 2). Only four out of 79 isolates of susceptible *Proteus mirabilis* were found to be resistant to nitroxoline; none of the 11 MDR *P. mirabilis* isolates expressed resistance to nitroxoline. Out of the 94 susceptible *K. pneumoniae* isolates that were tested, only five strains presented resistance to nitroxoline. Additionally, 16 out of 50 MDR *K. pneumoniae* isolates were nitroxoline-resistant (Figure 2).

There was no significant difference between nitroxoline resistance in MDR vs. susceptible *P. mirabilis* and *E. coli* isolates (two-sided Fisher’s exact test, *p* > 0.999). In case of the *K. pneumoniae* urine isolates, however, the MDR strains were significantly more resistant to nitroxoline than the susceptible strains (two-sided Fisher’s exact test, *p* < 0.0001).

### 2.3. Parallel Resistance to Cotrimoxazole and 3rd Generation Cephalosporins, Ciprofloxacin and MDR Strains of E. coli

We also tested for parallel resistance to cotrimoxazole and other antibiotics in *E. coli*, the by far most frequently occurring uropathogen. We observed that 20% of the susceptible urine isolates were resistant to cotrimoxazole. Among strains resistant to 3rd generation cephalosporins and to ciprofloxacin, cotrimoxazole resistance was detected to a much greater extent (42 to 46%). The highest rate of cotrimoxazole resistance (60%) was detected in MDR *E. coli* isolates (Table 1).

### 2.4. Co-Localisation of Transferable Resistance to Cotrimoxazole, ESBL and Quinolones

To exemplarily characterise the mobile resistance gene pool and the genomic basis for multidrug resistance of *E. coli* urine isolates, the complete genome sequence of a randomly selected multiresistant *E. coli* strain EC394-330266 was determined. In silico typing indicated that this strain belongs to phylogroup B2 and can be allocated to sequence type ST131 and serotype O25:H4. Due to the presence of the characteristic fluoroquinolone resistance-conferring *gyrA* S83L/D87N and *parC* S80L/E84V substitutions in the chromosome, this strain is placed into multidrug-resistant clade C2/H30Rx of ST131 isolates. Besides the 5,068,795-bp chromosome, the genome is comprised of two plasmids: the large 128,428-bp MDR plasmid pEC394-330266-1 (pMLST profile: IncF RST F2:A1:B-) and a small 1550-bp Col(MG828)-like plasmid pEC394-330266-2. The MDR plasmid pEC393-33026601-1 contains a 38.5-kb resistance region coding for β-lactam (*bla*_CTX-M-15_, *bla*_OXA-1_), aminoglycoside (*aadA*5), macrolide (*mphA*-*mrx*-*mphR*), fluoroquinolone (*aac*(6′)-Ib-cr), phenicol (*catB*3-like), sulfonamide (*sul*1), and trimethoprim (*dfrA*17) resistances. This region has a unique, complex composite structure, which is composed of two class 1 integron-like structures and several copies of IS*26* and IS*6100*. The latter IS elements may contribute to the formation of different conjugative transposons and translocatable units. The MDR region of the plasmid can be described as two mirror-image regions flanking a central section containing a *bla*_OXA-1_ and an *aac*(6′)-Ib-cr determinant as well as a gene encoding a CatB3-like acetyltransferase (Figure 3a,b).

To generally assess the extent of co-location of genes conferring resistance to cotrimoxazole and other AMR determinants in *E. coli* and related enterobacteria, we screened publicly available complete genome and plasmid sequences for co-existence of the class 1 integron-associated *sul*1 and *dfrA*17 alleles with other AMR genes. Our database search identified 332 class 1 integron structures as part of 286 different plasmids, 36 chromosomes or 10 not further specified genomic locations, which were deposited in the NCBI database (as of 30 November 2020) (Appendix A; Figure 4).

Altogether, 108 different resistance determinants conferring resistance to 17 different classes of antibiotics were detected. The number of AMR determinants present in the individual chromosomes or plasmids ranged between 3 and 18. The genes conferring resistance to aminoglycosides, sulfonamides and trimethoprim were observed to occur in multiple copies per genome. Additionally, genes coding for macrolide,- β-lactam-, cephalosporin- or tetracycline resistance were most frequently detectable. Out of 332 database entries with cotrimoxazole resistance genes, 20 also coded for fosfomycin resistance, 17 for nitrofurantoin resistance and 12 for both fosfomycin, and nitrofurantoin resistance.

## 3. Discussion

### 3.1. Co-Localisation of Transferable Resistance to Cotrimoxazole, ESBL and Quinolones

Therapeutic antibiotic use is a risk factor for MDR UTI [41] and the growing prevalence of MDR bacteria isolated from the urinary tract is of considerable concern [2,3,4,5,7].

Reducing carbapenem use and exploring alternative treatment options for low-grade ESBL-producing uropathogens is highly needed to prevent the emergence of additional carbapenem-resistant enterobacterial uropathogens. Although carbapenems are the antibiotics of choice for the treatment of catheter-associated UTIs with ESBL-producing *Enterobacteriaceae*, carbapenem-sparing therapies should be appropriate for treating the development of MDR organisms by reducing inappropriate antibiotic use.

Only few alternatives for treatment, such as cotrimoxazole, nitrofurantoin or fosfomycin, have been discussed [7,21,22]. Nitroxoline is recommended for the calculcated therapy of uncomplicated UTIs in adults [24]. However, it also appears to be a promising candidate for the therapy of MDR *Enterobacterales*, as it is active against practically all strains of *E. coli* and most other tested enterobacterial uropathogens, although not against *Pseudomonas aeruginosa* [25,26]. Nitroxoline retains its antibacterial activity even against challenging bacteria such as 3rd generation cephalosporin- and/or quinolone-resistant enterobacteria [25,27,42].

It has recently been reported that, in a particular case, nitroxoline was the only drug fully active against two different MDR uropathogens found in the same patient [43]. In the annual survey of laboratory results presented in this study, we confirm that nitroxoline remains active against many uropathogens, which are otherwise resistant to the commonly used antibiotics (Figure 2). Nitroxoline is thought to chelate divalent cations, thereby interfering with transcription by inhibiting the RNA polymerase as well as interfering with outer membrane integrity, subsequently leading to bacterial death [44,45]. Additionally, nitroxoline induces the SOS response and may, therefore, cause DNA damage similar to the structurally related quinolone antibiotics [46]. Based on the available data, there is no parallel resistance between nitroxoline and the first-line drugs recommended for UTI treatment. Chromosomal mutations that increase expression of the MDR pump EmrAB-TolC can confer resistance to nitroxoline in vitro. These mutations, however, do not lead to cross-resistance to nitrofurantoin [47].

Bacteria which were resistant to β-lactams or to β-lactams and quinolones were, in a relatively high proportion, concomitantly resistant to cotrimoxazole (Table 1, Appendix A, Figure 4), which is still one of the most frequently prescribed antibiotics against UTI [21]. The high proportion (>20%) of cotrimoxazole-resistant strains among MDR uropathogens, shows that, in accordance with the statements of Concia et al. [21] and Wagenlehner et al. [24], cotrimoxazole is no longer appropriate for the calculated therapy or prophylaxis of UTIs. Furthermore, substantial side effects, such as acute kidney injury in patients taking renin-angiotensin system blockers or hyperkalaemia, which may cause fatal cardiac arrhythmia, must be taken into account when using cotrimoxazole in comparison with other antibiotics [48].

Recurrent UTIs can result from persistent uropathogens present in biofilm [27,31]. This is particularly true for patients with permanent intravesicular catheters or foreign bodies in the urinary tract. Hence, the long-term use of antibiotics is necessary to prevent or cure such infective foci. Nitroxoline can act on susceptible bacteria in biofilms [27,45]. Resistance against nitroxoline is not transferred by plasmids or associated with other mobile genetic elements. Nevertheless, we cannot exclude that relevant genes or other determinants affecting nitroxoline susceptibility may be transferable via mobile genetic elements or transduction. Currently, this drug seems to be the preferred choice for the treatment of biofilms in the urinary tract.

In principle, cotrimoxazole is also active against bacteria in biofilms [49], but because of the high prevalence of resistance to this drug in normal urinary tract isolates (Table 1) and particularly in MDR strains, cotrimoxazole seems to be less suitable. Data from in vitro studies indicate nitroxoline is more suitable than cotrimoxazole for the therapy and prophylaxis of uncomplicated UTIs with MDR uropathogens. Additionally, the pharmacological properties speak in favour of the use of nitroxoline for the therapy of uncomplicated UTIs, since the accumulation of nitroxoline in urine is high, whereas cotrimoxazole is systemically available and is excreted not only via urine, but also via the intestine, where it may trigger the selection of resistant *Enterobacterales* in the resident intestinal microbiota.

### 3.2. Parallel Resistance Due to Accumulation of Resistance Determinants on Mobile Genetic Elements

The parallel resistance in MDR nosocomial pathogens results from the presence of plasmids and other mobile genetic elements (MGEs) simultaneously carrying various resistance genes [50,51]. For example, IncF resistance plasmids, which are frequently found in ST131 isolates, have the ability to acquire various resistance genes and to rapidly disseminate among *Enterobacterales* [52]. ST131 is one of the most common MDR UPEC lineages worldwide [53]. It is, therefore, not surprising that the arbitrarily chosen multi-resistant UTI isolate from a 70-year-old patient analyzed in more detail in this study also belongs to ST131. This strain harbors an IncF MDR plasmid (Figure 3) and is thus likely to be selected during antibiotic therapy, especially in clinical settings where antibiotics are often used. Since several resistance genes can either be present on one plasmid, as in case of pEC394-330266-1, or co-exist in the genome of an individual bacterial strain (Figure 4), the use of any of the antimicrobial agents according to the current guidelines, including cotrimoxazole, will potentially select for MDR bacteria.

The parallel resistance to cotrimoxazole and other common antibiotics such as penicillin, 3rd generation cephalosporins and ciprofloxacin is substantial. This results from the acquisition and further development of MGEs such as plasmids, transposons and integrons, carrying several other resistance genes, for example, those coding for resistance to β-lactams and quinolones. As multiple copies of transposons and IS elements are often distributed in a bacterial genome, they can promote genomic plasticity and DNA rearrangements due to homologous recombination. Plasmid-mediated horizontal transfer of resistance genes enables the rapid and extensive spread of resistance phenotypes in a short time [40,51,54].

In *E. coli*, the cotrimoxazole resistance determinants, such as the *sul*1 and *dfrA*17 allelic variants found in pEC394-330266-1, are often part of class 1 integrons, which are the most widespread class of integrons among enterobacteria. In association with functional Tn*402*, class 1 integrons can change their location within the genome [40,55]. Connected with non-functional Tn*402* variants, class 1 integrons are frequently described as “clinical integrons” [56]. They are often part of plasmids or larger transposons, such as Tn*21* or Tn*1696*, and can therefore be disseminated [57]. Furthermore, many class 1 integrons exhibit a 3′ region, which is usually composed of a *qacE*Δ1 gene conferring resistance to quaternary ammonium compounds [58], followed by a *sul*1 gene and orf5 encoding resistance to sulfonamides and a protein of unknown function, respectively. Bacteria carrying integrons are often multidrug-resistant [59,60,61,62]. Their resistance profile does not solely mirror the antibiotic-resistance cassettes of the integron, but also the presence of other resistance genes, present on other MGEs in the genome. This way, many resistance genes can be co-selected [60,63]. The presence of high numbers of plasmids in integron-positive isolates corroborates that the joined action of multiple MGEs facilitates the mobilization and spread of resistance determinants [64].

*bla*_CTX-M-15_ variants and other co-transfered resistance genes can be found on conjugative plasmids, especially multireplicon IncFII plasmids with additional FIA/FIB replicons [65]. In ST131 isolates, these plasmids exhibit considerable variability in their structure and gene content [66]. As can also be seen from the resistance region of pEC394-330266-1, IS*26* plays a prominent role in the capture and spread of different AMR genes as well as in DNA rearrangements and homologous recombination in Gram-negative bacteria [67,68,69,70,71,72,73,74,75,76,77] (Appendix A). IS*26* can form mobile composite transposons with tandem arrays of antimicrobial resistance genes [73,78,79]. Together with class 1 integrons and transposons they can form complex resistance gene loci (CRL) and so-called translocatable units (TUs), which have intracellular transposition ability. When they are part of a larger conjugative MGE, this can then facilitate their intercellular transposition [80]. IS*26*-flanked combinations of multiple resistance determinants were described for TUs in different MDR *E. coli* isolates [81,82,83].

It has been convincingly shown that a policy to restrict the use of fluoroquinolones will reduce the prevalence of ESBL producers [84]. A rational strategy to hold back the epidemic spread of MDR bacteria would be to use primarily those antibiotics for UTI therapy, which are not required for the treatment of other infections and for which plasmid-based resistance genes are not yet common, such as fosfomycin and nitrofurantoin. The plasmid-encoded resistance to several antibiotics is not only responsible for therapeutic failure, but will also lead to the selection of MDR bacteria [85], when a patient is treated with any of the antibiotics specified in Appendix A. Therefore, it can be expected that the use of cotrimoxazole for therapy, and particularly for the long-term prevention of recurrent UTIs, will select these MDR uropathogens [86]. Hence, it is obvious that the phenomenon of multi-resistant bacteria will become even more relevant during prophylaxis with cotrimoxazole. Our screen for the co-existence of AMR determinants in the 332 class 1 integron-positive *enterobacterial* genomes and plasmids demonstrated that, in addition to cotrimoxazole, some strains also already carry fosfomycin and/or nitrofurantoin resistance determinants.

## 4. Materials and Methods

### 4.1. Urine Culture Analysis and Antibiotic Susceptibility Testing

Urine samples from inpatients, as well as outpatients, of all ages with suspected UTI that were sent from practitioners as well as from different regional clinics to Laboratory Limbach/Heidelberg for microbiological examination in 2018. These samples were processed according to the German guidelines [87]. If the samples displayed bacterial counts exceeding 10^4^/mL and bore no more than three different pathogens in these urine samples, then the urine isolates were declared as relevant and included in the collection. Bacterial identification was generally performed by MALDI-TOF (Bruker, Bremen, Germany) and occasionally by VITEK 2 (Biomérieux, Nürtingen). The antibiograms for most antibiotics were also determined by VITEK 2. Nitroxoline is not included in the Vitek 2 AST cards. Susceptibility to nitroxoline (30 µg) has, therefore, only been tested upon the request of the sender by separate disk diffusion assays (Kirby–Bauer method). The interpretation followed the specifications of the European Committee on Antimicrobial Susceptibility Testing (EUCAST), i.e., susceptible: zone diameter > 15 mm corresponding to a MIC < 16 mg/L [26]. In our study, MDR bacteria were defined as being resistant to piperacillin, 3rd generation cephalosporins, carbapenems and quinolones. In case of carbapenem resistance, the presence of the following most prevalent carbapenemase classes, such as OXA-48, NDM-1, KPC, VIM, and IMP was tested.

### 4.2. Genome Sequencing

The genome sequence of multiresistant *E. coli* strain EC394-330266 isolated from the urine of a 70-year-old outpatient was analyzed in detail. This isolate was arbitrarily chosen from the collection of MDR *E. coli* urine isolates included into the resistance screening in order to assess and exemplarily describe the genomic basis for the parallel resistance of *E. coli* urine isolates to many antibiotics. Total genomic DNA was isolated with the MagAttract^®^ HMW DNA kit (Qiagen, Hilden, Germany). We combined long-read Oxford Nanopore MinION and short-read Illumina MiSeq platforms in order to completely close the genome sequence. We used the Nextera XT DNA Library Preparation kit (Illumina, Eindhoven, The Netherlands) to generate a 500 bp paired-end library, which was sequenced on the Illumina MiSeq sequencing platform with v2 sequencing chemistry. The paired FASTQ files were base called from the Illumina raw sequence read data. The MinION sequencing library was prepared with the Nanopore Ligation Sequencing Kit (Oxford Nanopore, Oxford, UK) and sequenced with a R9.4.1 MinION flow cell (24 h run using MinKNOW v2.0 (Oxford Nanopore, Oxford, UK) with the default settings). We used Guppy v3.3.0 (Oxford Nanopore, Oxford, UK) for base calling of the FAST5 files and conversion to FASTQ files.

FastQC (v0.11.5) (http://www.bioinformatics.babraham.ac.uk/projects/fastqc (accessed on 2 November 2020)) was used for the quality check of the raw reads. Sickle (v1.33) (https://github.com/najoshi/sickle (accessed on 2 November 2020)) was used for trimming low-quality reads. Long reads were filtered by quality by Filtlong v0.2.0 (https://github.com/rrwick/Filtlong (accessed on 2 November 2020)), with parameters: —min_length 1000 –keep_percent 90- target_bases 500000000)., Unicycler v0.4.8 [88] (https://github.com/rrwick/Unicycler (accessed on 2 November 2020)) with default parameters was employed for hybrid de novo genome assembly of the Nanopore and Illumina reads.

### 4.3. Annotation and Strain Typing

The complete genome sequence was annotated using the NCBI Prokaryotic Genomes Annotation Pipeline (PGAP) (v1.12) [89]. Plasmid types, serotypes, and acquired antibiotic resistance mechanisms we determined using PlasmidFinder (v2.0.1) [90], SerotypeFinder (v2.0.1) [91] and ResFinder (v4.1) [92], respectively, by a minimum sequence identity of 90% and a minimum sequence length of 80% to respective database entries. MobileElementFinder (v1.0.3) [93] was employed to identify mobile genetic elements and their insertion sites in pEC394-330266-1. The assembled genome sequence of *E. coli* isolate EC394-330266 01 was deposited at NCBI GenBank under the Bioproject number PRJNA640606.

### 4.4. Analysis of Co-Occurrence of Cotrimoxazole and Other Antimicrobial Resistance Genes in Bacterial Genomes

Publicly available bacterial genome sequences were screened for the presence of the *sul1* and *dfrA17* genes found on pEC394-330266-1 and other antibiotic resistance determinants. As *sul1* and *dfrA17* are frequently carried on class 1 integrons, the Integron Finder package v.1.5.1 was used to identify the integrons occurring in the 332 selected plasmid and chromosomal sequences [94]. Antibiotic resistance genes were identified using abricate v1.0.0 (https://github.com/tseemann/abricate (accessed on 2 November 2020)) using the NCBI AMRFinderPlus database with a nucleotide identity of >90% and coverage of >90%. The presence–absence profile of the antibiotic resistance genes either globally detected in the 332 genome sequences or in association with class 1 integrons and their flanking 40-kb sequence context was visualized.

## 5. Conclusions

Our data conclusively show that treatment of uncomplicated UTI according to the current guidelines bears the risk of promoting MDR uropathogens. Against this background, the low prevalence of nitroxoline resistance among enterobacterial uropathogens argues that nitroxoline remains a reasonable alternative for the treatment of such infections. Nevertheless, the future development of local and global nitroxoline resistance rates in uropathogens, as well as the molecular mechanisms involved in decreased nitroxoline susceptibility, will have to be carefully monitored. Likewise, we need to investigate the mechanisms and, where appropriate, the associated mobile genetic elements involved in the spread of nitroxoline resistance.

## Figures and Tables

**Figure 1 antibiotics-10-00645-f001:**
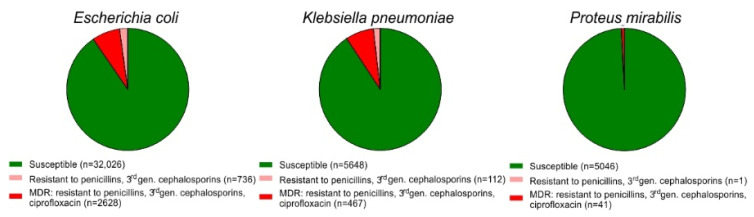
Incidence of resistance in 35,390 *E. coli*, 6227 *K. pneumoniae*, and 5088 *P. mirabilis* urine strains isolated in 2018 (green, susceptible; red, MDR, i.e., resistant against penicillins, 3rd generation cephalosporins and ciprofloxacin; light red, resistant against penicillins and 3rd generation cephalosporins).

**Figure 2 antibiotics-10-00645-f002:**
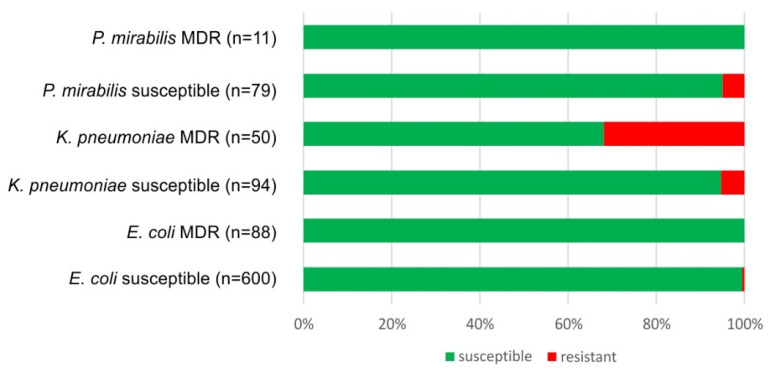
Susceptibility or resistance to nitroxoline among 922 urine isolates of various *Enterobacterales* (green, susceptible; red, resistant).

**Figure 3 antibiotics-10-00645-f003:**
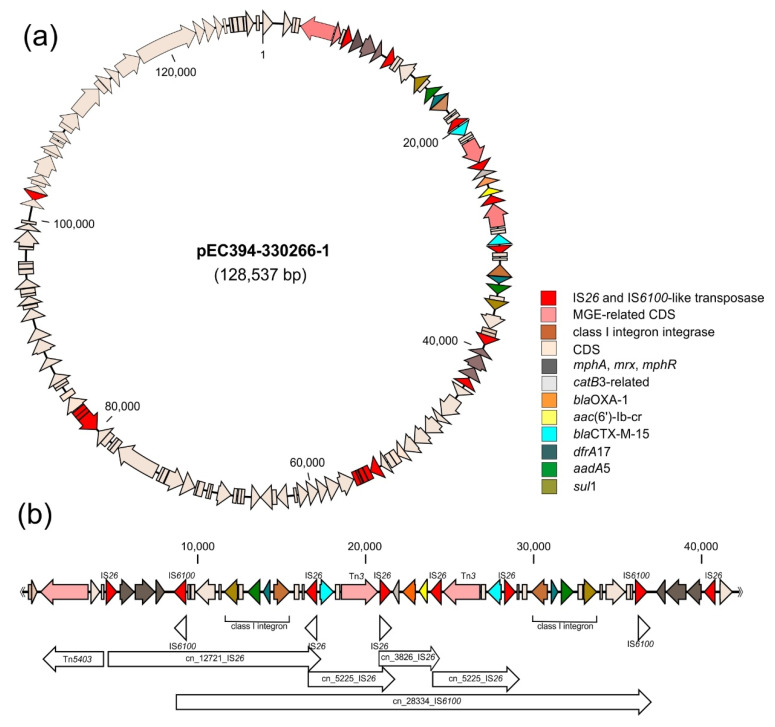
Multiresistance plasmid pEC394-330266-1 present in *E. coli* isolate EC394-330266. (**a**) Plasmid map (resistance genes and MGE-related genes have been highlighted in different colours). (**b**) Detailed structure of the MDR region of pEC394-330266-1. The different antibiotic resistance genes and multiple MGE-related genes have been indicated. The location of predicted class 1 integron structures, IS elements and conjugative transposons are given in the lower part of the figure.

**Figure 4 antibiotics-10-00645-f004:**
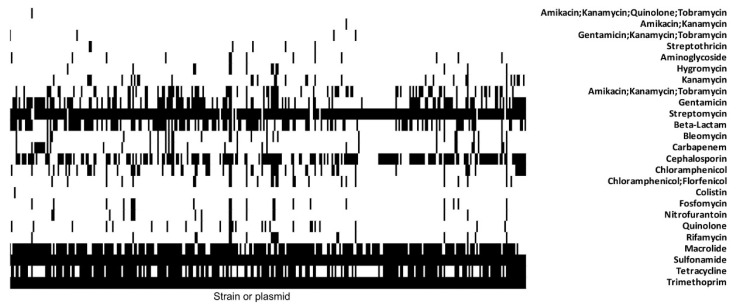
Presence-absence matrix to visualize parallel resistance in 332 enterobacterial isolates. The presence of resistance determinants conferring resistance to 17 groups of antibiotics was detected in silico (**black**, resistance gene is present; **white**, resistance gene is absent).

**Table 1 antibiotics-10-00645-t001:** Parallel resistance to cotrimoxazole and third generation cephalosporins or ciprofloxacin, respectively, in MDR *E. coli* urine isolates.

*E. coli* Urine Isolates	Cotrimoxazole Resistant Isolates [%]
Susceptible (*n* = 29,094)	20
3rd generation cephalosporin resistant (*n* = 763)	42
Ciprofloxacin resistant (*n* = 4450)	46
MDR (*n* = 2624)	60

## Data Availability

Data available in a publicly accessible repository that does not issue DOIs. Publicly available datasets were analyzed in this study. The assembled genome sequence of *E. coli* isolate EC394-330266 01 was deposited at NCBI GenBank under the Bioproject number PRJNA640606. The NCBI GenBank accession numbers of the publicly available bacterial genome sequences screened in this study are listed in Appendix A.

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
