# Peer review of "Compared with Cotrimoxazole Nitroxoline Seems to Be a Better Option for the Treatment and Prophylaxis of Urinary Tract Infections Caused by Multidrug-Resistant Uropathogens: An In Vitro Study"

_antibiotics, 2021, doi:10.3390/antibiotics10060645_

Round 1

Reviewer 1 Report

Dobrindt et al. provide an interesting, informative study on the rate of nitroxoline resistance in uropathogens and occurrence of these genes compared to those contributing to other antibiotic resistances. Overall, the script reads satisfactorily, but key pieces of information should be enhanced for clarity:

Overall: the authors need to go through the script and make sure bacterial names and genes are italicized.

L28: Please provide a brief description of multi-drug resistance (MDR) (e.g., how many drugs must the pathogen be resistant to? Do these need to be from different classes?)

L34-35: Please clarify “inappropriate initial antibiotic therapy”. Does this mean that patients being treated with UTIs are given the inappropriate antibiotics? Are antibiotics given prophylactically for UTIs?

L44: Please briefly describe this mechanism/target receptors?

L46: Second choice = last resort?

L46-47: What does local resistance pattern mean?

L48-49: “broad spectrum of different bacteria” – does this mean it is a broad spectrum antibiotic and effective against Gram positive/negative microbes?

L50-51: Provide a reference here.

L56-57: What implications does this have if detected in the urine? Does this mean the treatment was effective?

L59: Surely there are side effects to be discussed? Otherwise, why would this antibiotic not be more widely used?

L63-64: Provide some discussion on bacteriophages, which are also contributors to dissemination of antibiotic resistance.

L69-71: It is possible that exposure to one antibiotic can promote expression of an unrelated ARG, but how does this exposure promote the dissemination of unrelated ARGs?

L86-89: What is the rationale for testing for resistance to these antibiotics specifically?

L97: Why were only 600 E.coli isolates tested? Wasn’t the sample size >35,000 isolates? And, why were only 88 MDR E.coli isolates tested when the authors identified 2,628 MDR isolates? Similarly, I am not sure why Proteus mirabilis nor K. pneumoniae were tested as it was not stated that they were identified earlier in the script. This is a big area of confusion in the manuscript currently. The authors should include a table that identifies the different bacteria that were identified in the urine samples.

L97: Using the term “normal” to describe susceptible strains is unconventional and may be defined subjectively. The authors should change the wording to “susceptible” instead.

L118: Why was this particular strain chosen for genomic analysis?

Figure 4: Indicate the difference between the bolded letters vs. regular letters on the right.

L185-186: Why is this obvious? It is not obvious to me. Also, what are “common drugs”?

L196-197: What side effects can occur?

L198: Can a biofilm form in the urinary tract (without the presence of a catheter or foreign body)?

L201: Aren’t other antibiotics also able to act on susceptible bacteria in biofilms? Why is only nitroxoline discussed?

L202-203: Where is the evidence that this resistance is not transferred via plasmids? Could this be a possibility in the future?

L228-236: Authors should include some discussion around transduction as this is lacking.

L287: What does EUCAST stand for? Was antibiotic susceptibility testing done using replicates?

Conclusions: Conclusions section can be considerably expanded upon especially as there were a lot of data in the series of experiments conducted. Authors should also consider a future directions/experiments that could be conducted to determine rate of transmission of nitroxoline in bacterial populations (e.g., conjugation experiments).

Reviewer 2 Report

The manuscript "Nitroxoline or cotrimoxazole for the therapy and prophylaxis of urinary tract infections with multi drug resistant uropathogens?" consider phenotypic data from 35.390 E. coli from UTIs as well as some isolates of other Gram negative bacterial species from UTIs. They expanded the resistance testing by adding  testing for nitroxoline for a selection of MDR isolates. The results of these show low prevalence of nitroxoline resistance, which is useful information. 

One E coli MDR isolate was picked randomly for WGS and co-localization of transferable resistance to cotrimoxazole, ESBL and quinolone described. They further looked for the co-localization of Class 1 integrons, sul1 and dfrA17 with other AMR genes in publicly available complete genome and plasmid sequences. 

I have several concerns regarding the approach and conclusions in this paper. 

1) The paper starts by describing screening data, however these are not appropriately described. The collection and choice of UTI isolates other than E. coli is lacking.

The phenotypic data is not discussed at all against what is described otherwise in Europe and the world. 

2) No statistical analysis was done on the phenotypic data. It would be of interest to know if there was a significant difference between Nitroxoline resistance in MDR vs. sensitive isolates, as well as for the data in table 1. 

3) The random choice of a isolate for sequencing is not valid when the authors present this as a "typical" UTI isolate. There is plenty data in the literature which could be used to pick an isolate to better represent a typical MDR UTI isolate. The data from the sequencing is interesting, but can not be used in the maner presented here. 

4) The introduction and discussion should be shortened. The authors spend allot of space expanding on points which they do not connect to the current study. 

5) The references are given in different styles in the document. 

6) The authors do not follow the common nomenclature for bacterial names (should be given in italics).

7)I find it unlikely that the authors conducted the phenotypic resistance screen described here. It is also not mentioned in the Acknowledgements.

Round 2

Reviewer 1 Report

Figure 1 legends and text (L100-102) do not match. For instance, in L102, the authors state that E.coli is resistant to quinolones, but in the caption it states only ciprofloxacin.

Author Response

As suggested, we have replaced in line 100 “quinolones” by “ciprofloxacin”. We also tried to improve the quality of the English language used in the manuscript.